# Evaluating Local Multilingual Health Care Information Environments on the Internet: A Pilot Study

**DOI:** 10.3390/ijerph18136836

**Published:** 2021-06-25

**Authors:** Russell Miller, Nicholas Doria-Anderson, Akira Shibanuma, Jennifer Lisa Sakamoto, Aya Yumino, Masamine Jimba

**Affiliations:** 1Department of Community and Global Health, Graduate School of Medicine, The University of Tokyo, Tokyo 113-0033, Japan; rmille0110@gmail.com (R.M.); nicholas.doriaanderson@gmail.com (N.D.-A.); jennylisa.sakamoto@gmail.com (J.L.S.); 2cor1130@gmail.com (A.Y.); mjimba@m.u-tokyo.ac.jp (M.J.); 2Kawasaki Health Cooperative Association, Asao Clinic, Kawasaki 210-0833, Japan

**Keywords:** information dissemination, communication barriers, health resources, Japan

## Abstract

For foreign-born populations, difficulty in finding health care information in their primary language is a structural barrier to accessing timely health care. While such information may be available at a national level, it may not always be relevant or appropriate to the living situations of these people. Our objective was to explore the quality of online multilingual health information environments by pilot-testing a framework for assessing such information at the prefectural level in Japan. The framework consisted of five health care domains (health system, hospitals, emergency services, medical interpreters, and health insurance). Framework scores varied considerably among prefectures; many resources were machine-translated. These scores were significantly associated with foreign population proportion and the number of hospitals in each prefecture. Our multilingual health care information environment (MHCIE) framework provides a measure of health access inclusivity, which has not been quantified before. It is adaptable to other international contexts, but further validation is required.

## 1. Introduction

With the growth of the Internet, more people are making decisions about accessing health care using information they were able to acquire online. Online health information-seeking is often assumed to be related only to finding quality information about specific health conditions [1,2,3]. Internet-based health care information, in contrast, is designed to help consumers navigate their environment to understand the health care options available to them locally [4,5].

Multilingual information about local health care is particularly important for supporting the health of foreign-born populations (like short-term tourists [6] and long-term migrants [7]) who may be navigating unfamiliar geographies and languages [8,9]. Despite the importance of this data, no frameworks exist to evaluate geographically localized environments of Internet-based health care information [10]. 

In non-English-speaking host countries, language barriers may be even more severe for foreign-born populations when seeking health information. Non-English language barriers limit the number of resources in either their native language or English [11]. The COVID-19 pandemic has given new importance to global health initiatives, like strengthening health information systems to promote healthy migration and to protect greater population health [12].

Such global health initiatives are needed to address structural, linguistic barriers to health that persist in non-English-speaking countries like Japan. Even with universal health coverage [13] and essential medical care guaranteed, if patients can cover the associated costs [14], the universal access paradigm has been complicated by hosting 31,882,000 overseas tourists [15] and 2829,400 foreign residents [16] in 2019. Since the migration-demographic transition of Japan [17], research is emerging that foreign-born resident populations face serious health disparities in terms of access to health care [18,19]. Access to translated health care information is necessary to ensure a living environment inclusive of inbound migrants and the human right to health. 

Our research question was: How robust are offerings of online multilingual health care information for information-seekers in non-English speaking countries? Thus, our objective was to explore the quality of online multilingual health information environments by pilot-testing a framework for assessing such information. This framework has the potential to provide actionable evidence from which to target inclusive health care policy.

## 2. Methods

### 2.1. Study Design

We systematically searched English-language health care information that was available on websites local to the 47 prefectures of Japan. We chose a prefecture as the unit of evaluation because prefectural governments oversee local health systems in Japan. Each prefectural collection of websites represented a local multilingual health care information environment. In the next stage we identified appropriate domains and scoring levels from the search results to develop a framework that could standardize the variability observed across environments. Finally, we pilot-tested the framework to assess its performance and visualize disparities at the prefectural level.

### 2.2. Search Strategy

Our search strategy was designed as a real-world sampling of multilingual health care information from a search engine. This means that the search was not artificially exhaustive but mimicked exploratory searches, in each prefecture, that a health care information-seeker would likely use [20,21]. We sought information sources using the Google search engine commonly utilized for health information-seeking searches by English-speaking populations [22]. The researchers’ computers were localized to IP addresses in Japan. 

Search strings were formulated based on a review of recent publications documenting the health care challenges of migrants in Japan: accessing health insurance [23], cancer screening [19], COVID-19 treatment [24], cross-cultural maternal health care [25], HIV-testing [14,26], and reproductive health information [27]. Six search strings were used, each prefecture’s name (*n* = 47) and one of the following health care information terms: “health system”, “hospital list”, “hospitals”, “emergency services”, “medical interpreters”, or “national health insurance.” These 282 search string combinations mimicked typical search query syntax [28]. The first three pages of search results (30 links) were assessed for inclusion with each search [29]; 8460 links were reviewed in total. 

The inclusion criteria for information sources were: (1) websites that presented information about the health care covered by any of the search terms (not necessarily the specific term that was used to generate the result); (2) presentation of that information was at least partially in English; and (3) the website was administrated by an organization local to the prefecture of interest (e.g., governments, medical organizations and other non-profit organizations).

All searching was completed during March 2020.

### 2.3. Data Extraction

The data extraction sheet was grouped by search term, and the data fields were the organization administering the website, webpage title, web address, brief description, and notes on readability/translation (if poor) (Appendix A). Two researchers (R.M. and N.D.-A.) reviewed the search results and extracted data from web pages that met the inclusion criteria.

### 2.4. Developing a Scoring Framework for Multilingual Health Care Information Environments

Based on content analysis of the health care information in the search results, five-item domains were defined: information on the overall health system, on hospitals, on emergency health services, on medical interpreters and on health insurance. Each domain was scored with one item, except the hospital domain, which had two items: one for general and institution-specific information, the other for the number of hospital pages detected in the environment. Furthermore, specialty health information, information not described elsewhere in the results, was added as a seventh item to be more comprehensive. 

The six main items were scored on an unweighted scale from 0 to 3 points (0—not present/poor; 1—somewhat satisfactory; 2—satisfactory; 3—excellent). Each scoring level was given further detailed criteria based on the same content analysis as the search results. Environments that had specialty health information were awarded 2 extra points, which expanded the maximum score to 20. The final scoring assessment for pilot testing in Japan was called the Multilingual Health Care Information Environment (MHCIE) framework (Table 1).

The MHCIE framework was inspired by the flexibility of the QUality Evaluation Scoring Tool (QUEST) that assesses the quality of online articles about health disorders [30]. Like QUEST, we created seven scoring items with item scores ranging from 0 to 3. However, unlike QUEST, our framework aimed to holistically integrate and assess multiple information sources on health care.

For scoring purposes in the pilot study, prefectures were split into two groups for scoring a priori because foreign residents and visitor populations have historically concentrated in a few Japanese prefectures. A distinct gap in information environments was expected between the two groups. Group 1 (*n* = 6) included only prefectures with populations among the top 12 of all prefectures for both foreign resident population and foreign tourist overnight stays [16,31]. To prevent ceiling effects that would skew the average and regional scores Group 1 prefectures had heightened criteria in terms of quality and number of available information sources. Group 2 was comprised of the remaining prefectures (*n* = 41).

The same researchers independently scored the information environment from each prefecture. Scores with a discrepancy greater than 2 points between researchers were reconciled through consultation. The remaining, discrepant scores were resolved by averaging both researchers’ scores with scoring by a third researcher (J.L.S) who was naïve to the study. Average prefecture scores were rounded to the nearest whole number and were then used to calculate regional scores and populate a nationwide score heat map. We also mapped hospitals with multilingual webpages which were captured in our search.

### 2.5. Statistical Analysis

The normality of the prefecture grouping was checked. Then, Group 1 scores were ungrouped to Group 2 criteria for integrated analyses of rubric scores and several prefectural factors to assess the relationship between general prefectural characteristics and health care information (Appendix A). The normality of score distribution was tested with skewness/kurtosis tests and the Shapiro–Wilk test. For further analysis nine explanatory variables were selected from four variable categories: scale of prefecture (total population, population density, financial power index), foreign residents and tourists (foreign resident population, foreign residents per 1000 prefecture population, ratio of tourists to prefecture population), medical facilities (number of hospitals, number of clinics, total number of medical facilities (hospitals + clinics), facilities per 1000 inhabitants), and unobserved fixed effects (Group 1 scores as a dummy variable).

Selection from each variable category to create a model was based on bivariate correlation analysis using heteroscedasticity-robust standard error to cope with homoscedasticity and normality assumptions. A multicollinearity check of the model was performed to better fit the model by eliminating inappropriate factors. A final ordinal least squares (OLS) regression model was checked with generalized linear modeling (GLM) and Bayesian regression modeling using the same explanatory variables. Post-estimation of residual normality was also executed. Statistical significance level was set at 5%. All statistical tests were performed with STATA 13.1.

### 2.6. Reporting

This report follows the Standards for QUality Improvement Reporting Excellence (SQUIRE) 2.0 reporting guidelines for systematic, data-driven efforts to improve the quality, safety and value of health care (Appendix A) [32].

## 3. Results

### 3.1. Adapting MHCIE to the Available Health Information Sources

Out of the cumulative 8460 search engine results from all search strings for all prefectures 517 were eligible for data extraction (hit rate: 6.1%). The mean (SD) number of hits for each prefecture was 11 (5.2), and the median (range) was 11 (3–27). 

A low hit rate in the Japanese context severely limited the assessment levels that would be appropriate for the MHCIE framework. For example, the lowest level of item criteria for Group 2 had to assume no eligible information could be found for each item. In addition, to cope with the paucity of relevant websites, the maximum level for number of hospitals was capped at ≥7. 

The MHCIE framework had an overall unadjusted inter-rater agreement of approximately 65% for the analysis of 47 local information environments. The scores for the remaining environments with wide ranges were averaged with scoring from a third reviewer to reach a qualitative consensus to reach 100%. It was concluded that the framework provided sufficient rigor and flexibility to justify further data analysis and statistical analysis as a basis for future research and development.

### 3.2. Pilot Study of Subnational-Level Information Environments

The nationwide average score was 12 out of a possible 20 points. Tohoku (8.3 points) in the northeast and Shikoku (6.3 points) in the southwest had the lowest average regional scores (Figure 1). Both regions consisted exclusively of Group 2 prefectures.

A summary of the score distributions resulting for each item of MHCIE is presented in Table 2. For prefectures in Group 1 the average score was 14 out of 20 points. Most (62.5%) data category scores for this group received 2 or 3 points. The majority of Group 1 had excellent information for the overall health system and emergency health services categories, but their resources such as hospital lists, hospital web pages, medical interpreters, and national health insurance categories scored 1 or 2 points. Only two prefectures, Osaka and Kyoto, had specialty health information available in English. 

For the remaining prefectures (Group 2), the average score was 11 out of 20 with two prefectures scoring less than 5, or ‘very limited’. There were seven prefectures (17.1%) with a score of 0 regarding information on the health system overall. Most prefectures had five or fewer hospital webpages in English (≤2 points), and there was no, or very limited, information on medical interpreters for about half of Group 2. Lastly, a minority of prefectures (18.3%) had specialty health information. 

Medical facilities with English pages captured during the search were mapped with the available languages (Figure 2). Nationwide, 123 unique hospital pages were extracted during the search which represents 1.5% of the 8442 hospitals in Japan. This total included 36 university hospitals (29.3%) and 9 clinics (7.3%); 38 (30.9%) pages were also translated into Simplified Chinese compared to less than 13% for Korean and other languages, respectively.

### 3.3. Statistical Analysis

In sensitivity analysis, the combined Group 2 and simulated Group 2 data (for Group 1 prefectures) were found to be strongly bimodal around the scores of 13 and 17, with a strong right-leaning skew. The higher mode was primarily populated by the augmented Group 1 prefectures. The skew and modality of the data suggested the split-group framework was justified and valid for use in the research.

All prefectural variables of interest were diagnosed as not divergent from a normal distribution and were considered roughly normally distributed (Shapiro–Wilk test; skewness = 0.767, kurtosis = 0.440). The final regression model after residuals check is presented in Table 3. After being controlled for population density, the ratio of tourists and fixed effects, the health care information score was associated with the number of foreign residents per 1000 inhabitants (B: 0.3; 95% CI: 0.1–0.5, *p* = 0.006) and the total number of hospitals (B: 1.6; 95% CI: 0.6–2.7, *p* = 0.003). These results were largely consistent with other modeling approaches.

## 4. Discussion

The MHCIE framework was able to provide qualitative and quantitative insights about disparities in the strength of web-based, multilingual health care information environments at a sub-national level. Once properly adapted, the framework coped well with the few, heterogeneous information sources found in the pilot study of Japan. Our results strongly suggest that less populated regions outside of large metropolitan areas struggle to provide a minimal amount of information non-native Japanese speakers can use to navigate the local health care system. There was a significant association between information scores and foreign resident population as well as with hospital count; it is likely that demographics were limiting the development of multilingual health care information. Similar to other countries, in order to attract the dynamism of tourists and migrants, more incentives and measurements may be required to improve the inclusivity of local health care.

In balancing between ease of use, concision and comprehensiveness, the MHCIE framework was relatively successful. The level of agreement among researchers could be considered appropriate to assess the framework’s feasibility for a pilot study [33]. Items in the framework were flexible enough to have a substantial inter-rater agreement; however, the items may benefit from more closed-ended criteria for more granular and consistent assessment. For example, a study assessing the usability of health information on medical institution websites (speed of information acquisition, the visual organization, and the discoverability via search engine) used a yes/no checklist which allowed for more consistency in terms of inter-rater reliability [34]. Future research should aim to assess the reliability and convergent validity of MHCIE in comparison to other health information tools [35].

Our framework could also be applied to any language and in other subnational health care environments. The framework’s generalizability is likely to be high because language or country should not fundamentally alter the relevance of any input criteria; an understanding of the local availability of multilingual health information may be required to adapt per-score levels for each category. As was noted in this pilot study, it was common for translated information to appear in English but not in Simplified Chinese or Korean. If scores in these languages were assessed, they would be much poorer than English, as has been documented in other multilingual health information research (French, German, Spanish) [36]. Using English as the anchor assessment language may be a viable option when using this framework. Comparing MHCIE scores between languages in the same locality and comparing to the distributions of nationalities of the foreign-born population would be another insightful usage.

Variability between multilingual information environments was related to the number of local foreign residents and medical facilities. While not unexpected, one reason for variability may be because web-based health seeking is relatively uncommon in Japan [37] and therefore less prioritized in non-cosmopolitan areas than in foreign countries [38]. Another reason for variability may be that Japanese rural areas have issues with health care accessibility even in the primary language, due to low population density [39]. Conversely, in locales with a large metropolitan foreign-born population, 90.7% of local government websites were translated into at least one foreign language [40]. Similarly, US hospitals with higher bed numbers, higher revenue, and more admissions were positively associated with providing non-English language services information [41]. Only with short-sighted policy is multilingual health care information being translated, post-hoc, to support a current foreign-born constituency rather than to attract or prepare for new foreign populations. 

Access to useable information in minority languages remains an issue to inclusive health care. The prevalence of machine translation across all information categories was high in Japan but a minority of translated information seemed to be quality-checked. This same problem has also been observed with Chinese public health materials in the US [42]. We observed that small municipal organizations independently disseminated some multilingual health care navigation including translated guides for daily life. In Japan, medical communication strategies such as ‘easy Japanese’ [43], a hospital accreditation system for accepting foreign patients [44], and calls for more medical interpreters [45] also deserve more action from policy-makers. 

Internationally, increasing the availability and quality of multilingual health care information is contingent on empowering quality-control measures for local and national governments after determining their specific needs. A potential strategy for creating these measures could be constructed based on mimicking behaviors from localities, which are detected to exhibit a positive deviance in scores by the assessment framework. In line with recent calls for inclusion of migrants in health information strategies [46], incentivizing prefectural and municipal governments to create and maintain quality, multilingual information throughout the health-seeking process should be a priority.

### Limitations

There are several limitations to this study that should be noted. The scope of the health care information search was limited for useability and did not cover all possible multilingual health materials. This shortcoming was partially mitigated by including all eligible health information resources identified within any of the search terms. Furthermore, Chinese and South Korean nationals comprise the majority of migrants and tourists to Japan and likely speak a first language other than English. While English remains the most widely used across multiple nationalities, further language assessments are warranted.

We chose prefectures as our point of investigation for this study because of their impact on geopolitically bound health systems and because demographic and structural data were available at this level. Prefectures are large administrative units, and they could obscure smaller populations or geographies which are poorly served within an otherwise outstanding health system. Second, the natural point of intervention based on these data would be at the prefectural level, promoting a top-down public health approach, which might not be flexible or specific enough to care for individual groups. Both problems might be avoided by successive testing within each prefecture at the district or municipality level. More localized testing would prompt adaptation of the framework to these units.

Furthermore, search engine algorithms are designed to suggest future search results based on previous web traffic. It is possible that search results were biased to suggest relevant resources because of the website cookies accumulated during the sequential examination. For this reason, the first prefecture searched, Hokkaido, was also the last prefecture searched (as a redundant search) to more adequately normalize every prefecture’s results. These resource collections should be considered a representative, yet non-exhaustive, sample of available health care information.

Any attempts at adapting the MHCIE framework to other contexts should be done considering vast cultural and political differences. For example, information on the National Health Insurance (NHI) scheme would have to be heavily adapted or removed entirely for contexts like the United States, which lacks universal health coverage or other resource poor settings. The framework should be able to accept any changes made to account for these contextual shifts, which can be considered a strength.

It should also be noted that the MHCIE framework has not been validated but was designed to assess the feasibility of understanding existing online environments for minority language health care specific to Japan. Future research should externally validate this framework and provide systematic guidelines for adaptability and flexibility across locales.

## 5. Conclusions

Multilingual health care information environments in a single bounded geography can be difficult to compare to each other in a standardized way but such measurement is necessary to know where interventions are required to improve such environments. Instead of assessing information resources individually, the MHCIE framework provided a more generalizable comparison. In this pilot analysis of subnational information in Japan there was considerable heterogeneity but with a framework they could be assessed to provide meaningful conclusions and targets for inclusive health policy.

## Figures and Tables

**Figure 1 ijerph-18-06836-f001:**
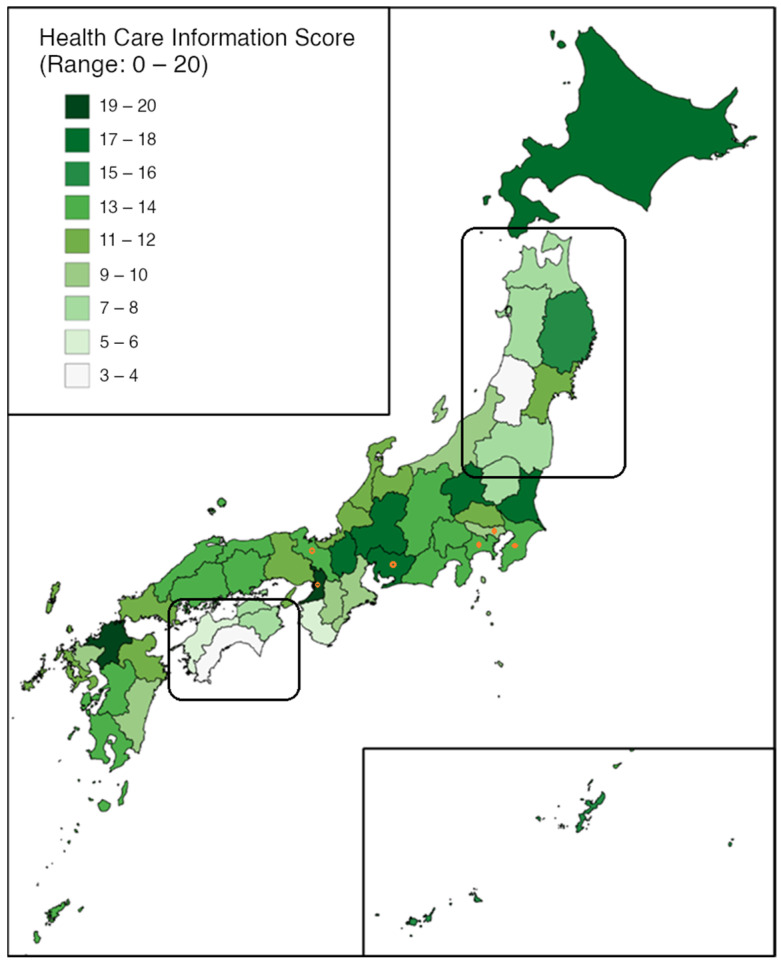
A geographic heatmap of multilingual health information environments by Japanese prefecture (roughly north-south orientation). Low scoring regions are outlined with rounded-edge boxes. Group 1 prefectures with a large foreign national population are marked with orange dots.

**Figure 2 ijerph-18-06836-f002:**
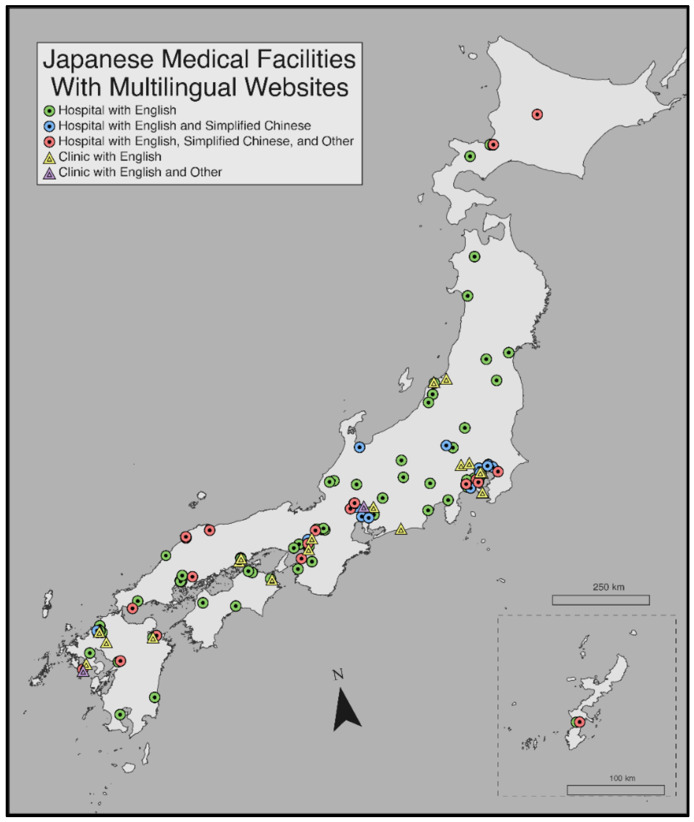
Map of health care facilities with multilingual webpages (*n* = 123) according to the real-world search results.

**Table 1 ijerph-18-06836-t001:** MHCIE framework adapted for use in assessing information environments.

Group 1 Prefecture Name:						
	0 points	1 point	2 points	3 points		
	**No/Limited information** (level spilt into two levels for Group 2) - Out of date - Machine translation - Few resources - Poor readability (1–2 hospitals pages in target language)	**Some information** - Mixed quality - Too much excess information - Adequate number of resources (3–5 hospitals pages in target language)	**Enough information** - Good quality - Comprehensive (5–7 hospitals pages in target language)	**Excellent information** (level not included for Group 2) - High-quality - Comprehensive - Resources are easy to use (≥7 hospitals pages in target language)		
1. Overall health system information						
2a. Information on hospitals						
2b. Number of professional English pages for hospitals						
3. Information on emergency health services						
4. Information on medical interpreters					Extra +2 *	(Min = 0, Max = 20)
5. Information on national health insurance						Total score
**Scores**						

* Specialty Information were awarded to prefectures which demonstrated additional capacities for foreign language health care services not explicitly described in the grading rubric. Examples of additional capacities include: disability services, vaccinations, emergency notification systems, HIV/STI testing and treatment, rehabilitation and addiction recovery, neonatal/infant/child/geriatric specialized care, or access to in-formation (health systems, hospitals, emergency, interpreter, NHI) in languages other than English.

**Table 2 ijerph-18-06836-t002:** A summary MHCIE score table for Japan.

Group 1 (*n* = 6)			Group 2 (*n* = 41)		
Overall Average Score		14	Overall Average Score		11
Overall score distribution			Overall score distribution		
0–5 = Limited		0	0–5 = Very limited		2 (4.9)
6–10 = Some		1 (16.7)	6–10 = Limited		13 (31.7)
11–15 = Enough		3 (50.0)	11–15 = Some		20 (48.8)
16–20 = Excellent		2 (33.3)	16–20 = Enough		7 (17.1)
	Reviewer 1	Reviewer 2		Reviewer 1	Reviewer 2
	*n* (%)	*n* (%)		*n* (%)	*n* (%)
Health system overall					
0—None or limited	0 (0)	1 (16.7)	0—None or very limited	7 (17.1)	7 (17.1)
1—Some	1 (16.7)	0 (0)	1—Limited	4 (9.8)	7 (17.1)
2—Enough	0 (0)	1 (16.7)	2—Some	9 (22.0)	12 (29.3)
3—Excellent	5 (83.3)	4 (66.7)	3—Enough	20 (48.8)	15 (36.6)
Hospital lists					
0—None or limited	0 (0)	0 (0)	0—None or very limited	1 (2.4)	0 (0)
1—Some	0 (0)	3 (50.0)	1—Limited	7 (17.1)	5 (12.2)
2—Enough	2 (33.3)	1 (16.7)	2—Some	14 (34.1)	19 (46.3)
3—Excellent	4 (66.7)	2 (33.3)	3—Enough	18 (43.9)	17 (41.5)
Number of hospitals with quality English webpages					
0—0–2 Hospitals	1 (16.7)	0 (0)	0—0 Hospitals	7 (17.1)	1 (2.4)
1—3–5 Hospitals	1 (16.7)	1 (16.7)	1—1–2 Hospitals	19 (46.3)	11 (26.8)
2—5–7 Hospitals	2 (33.3)	2 (33.3)	2—3–5 Hospitals	9 (22.0)	19 (46.3)
3— > 7 Hospitals	2 (33.3)	3 (50.0)	3—>5 Hospitals	5 (12.2)	10 (24.4)
Information on emergency health services					
0—None or limited	0 (0)	1 (16.7)	0—None or very limited	8 (19.5)	7 (17.1)
1—Some	1 (16.7)	0 (0)	1—Limited	4 (9.8)	8 (19.5)
2—Enough	1 (16.7)	2 (33.3)	2—Some	13 (31.7)	8 (19.5)
3—Excellent	4 (66.7)	3 (50.0)	3—Enough	15 (36.6)	18 (43.9)
Information on medical interpreters					
0—None or limited	1 (16.7)	2 (33.3)	0—None or very limited	20 (48.8)	21 (51.2)
1—Some	1 (16.7)	2 (33.3)	1—Limited	3 (7.3)	5 (12.2)
2—Enough	0 (0)	0 (0)	2—Some	4 (9.8)	8 (19.5)
3—Excellent	4 (66.7)	2 (33.3)	3—Enough	13 (31.7)	7 (17.1)
Information on national health insurance					
0—None or limited	0 (0)	0 (0)	0—None or very limited	3 (7.3)	2 (4.9)
1—Some	0 (0)	1 (16.7)	1—Limited	5 (12.2)	8 (19.5)
2—Enough	3 (50.0)	3 (50.0)	2—Some	9 (22.0)	9 (22.0)
3—Excellent	3 (50.0)	2 (33.3)	3—Enough	23 (56.1)	22 (53.7)
Specialty information					
0—No	4 (66.7)	4 (66.7)	0—No	32 (78.0)	35 (85.4)
2—Yes	2 (33.3)	2 (33.3)	2—Yes	9 (22.0)	6 (14.6)

**Table 3 ijerph-18-06836-t003:** Factors associated with health care information score.

R-Squared	0.490	(95% CI)	*p*-Value
	B		
Population density	−1.6	(−3.4, 0.2)	0.075
Number of foreign residents per 1000 inhabitants	0.3	(0.1, 0.5)	0.006
Overnight tourist stays per population	1.0	(−0.4, 2.3)	0.165
Number of hospitals	1.6	(0.6, 2.7)	0.003
Financial power index	−4.1	(−17.4, 9.1)	0.535
Group 1 (fixed effects)	1.5	(−2.7, 5.8)	0.470

## Data Availability

All data relevant to the study are included in the article or uploaded as Appendix A. A curated resource list from data extraction (all information categories and hospital webpages) are publicly accessible an online data repository (https://osf.io/cmvn6/; accessed on 14 June 2021).

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
