# Peer review of "Evaluating Local Multilingual Health Care Information Environments on the Internet: A Pilot Study"

_ijerph, 2021, doi:10.3390/ijerph18136836_

Round 1
Reviewer 1 Report
All the concerns are well addressed. I would suggest to accept the paper in the current form.
Reviewer 2 Report
This is a much improved focus for the work and describes the subject clearly.
I commend the authors for taking the previous reviewers feedback on board and the paper is now in a much better place because of that.....
This manuscript is a resubmission of an earlier submission. The following is a list of the peer review reports and author responses from that submission.
Round 1
Reviewer 1 Report
This paper systematically assessed each of Japan’s 47 prefectures as a basis for evidence-based policy reform on inclusive healthcare practices. Generally speaking, this is a good paper. The research question is important and the paper is easy to follow. The methodology is reasonable and rigorous. The results are also relevant to practice.
Minor issue
Line 99 (the following contents should be deleted.):
- Results
This section may be divided by subheadings. It should provide a concise and precise description of the experimental results, their interpretation, as well as the experimental conclusions that can be drawn.
Reviewer 2 Report
This is fairly well presented and crafted paper and I have no negative comments to make about the way the research was conducted or its conclusions.
I am not persuaded that the paper delivers findings that are of particular, significant relevance or interest to the international online journal readership.
There may be more appropriate routes the authors can look to for dissemination.
Round 2
Reviewer 2 Report
I note the work that the authors have undertaken to respond to specific issues raised during the review process. This does not address my more generic concern which relates to the contribution to scientific knowledge and understanding.